# Rapid vertical exchange at fronts in the Northern Gulf of Mexico

Lixin Qu [1,2] ✉, Leif N. Thomas [1] ✉, Aaron F. Wienkers [3], Robert D. Hetland [2], Daijiro Kobashi[4], John R. Taylor[3], Fucent Hsuan Wei Hsu [5], Jennifer A. MacKinnon [6], R. Kipp Shearman[5] & Jonathan D. Nash[5]

Over the Texas-Louisiana Shelf in the Northern Gulf of Mexico, the eutrophic, fresh Mississippi/Atchafalaya river plume isolates saltier waters below, supporting the formation of bottom hypoxia in summer. The plume also generates strong density fronts, features of the circulation that are known pathways for the exchange of water between the ocean surface and the deep. Using high-resolution ocean observations and numerical simulations, we demonstrate how the summer land-sea breeze generates rapid vertical exchange at the plume fronts. We show that the interaction between the land-sea breeze and the fronts leads to convergence/divergence in the surface mixed layer, which further facilitates a slantwise circulation that subducts surface water along isopycnals into the interior and upwells bottom waters to the surface. This process causes significant vertical displacements of water parcels and creates a ventilation pathway for the bottom water in the northern Gulf. The ventilation of bottom water can bypass the stratification barrier associated with the Mississippi/Atchafalaya river plume and might impact the dynamics of the region's dead zone.

Human activities and climate change have escalated coastal vulnerability. Coastal problems, including sea level rise, extreme flooding, dead zones, oil spills, marine debris, and harmful algal blooms, have aroused public concern for the sustainability of coastal environments. The Texas-Louisiana shelf in the Northern Gulf of Mexico (Fig. 1a) faces all these challenges. Here in particular, excess nutrients from the Mississippi/Atchafalaya River, combined with the suppression of vertical mixing by the strong stratification associated with its freshwater plume, lead to the formation in the spring and summer of a dead zone over the shelf, where the concentration of dissolved oxygen can be so low that marine life can no longer be sustained[1,2]. The river plume also forms density fronts, features in the flow characterized by strong lateral density gradients, where isopycnals (surfaces of constant density) are tilted and can act as a conduit connecting the surface waters with the bottom waters. This raises the possibility that, instead of penetrating the stratification barrier of the plume via vertical mixing, slantwise vertical motions at these fronts can bypass the stratification barrier by transporting water along their inclined isopycnals and ventilate the oxygen-deficient bottom waters, potentially alleviating hypoxic conditions. In addition, such slantwise vertical motions could mix carbon and heat, with implications for ecosystems, fisheries, and ocean heat content, the latter of which is of noted importance in this hurricane-prone coast.

During the summer when the dead zone is most prominent on the Texas-Louisiana shelf, a diurnal land-sea breeze is notable[3], and the fronts are forced by the land-sea breeze, which can extend out to the shelf edge (Fig. 1a). Given the latitude of the shelf (near 30 °N), these diurnal winds are near-resonant with the local inertial frequency ($f = 2\Omega \sin\phi$, where $\Omega$ is the rotation rate of the Earth and $\phi$ is the latitude), and generate intense currents that oscillate at frequencies close to $f$[3], while the tides in this region are relatively weak[4]. This opens the possibility for

[1]Department of Earth System Science, Stanford University, Stanford, CA, USA. [2]Pacific Northwest National Laboratory, Richland, WA, USA. [3]Department of Applied Mathematics and Theoretical Physics, University of Cambridge, Cambridge, UK. [4]Department of Oceanography, Texas A&M University, College Station, TX, USA. [5]College of Earth, Ocean and Atmospheric Sciences, Oregon State University, Corvallis, OR, USA. [6]Scripps Institution of Oceanography, University of California San Diego, La Jolla, CA, USA. ✉e-mail: lixinqu123@gmail.com; leift@stanford.edu

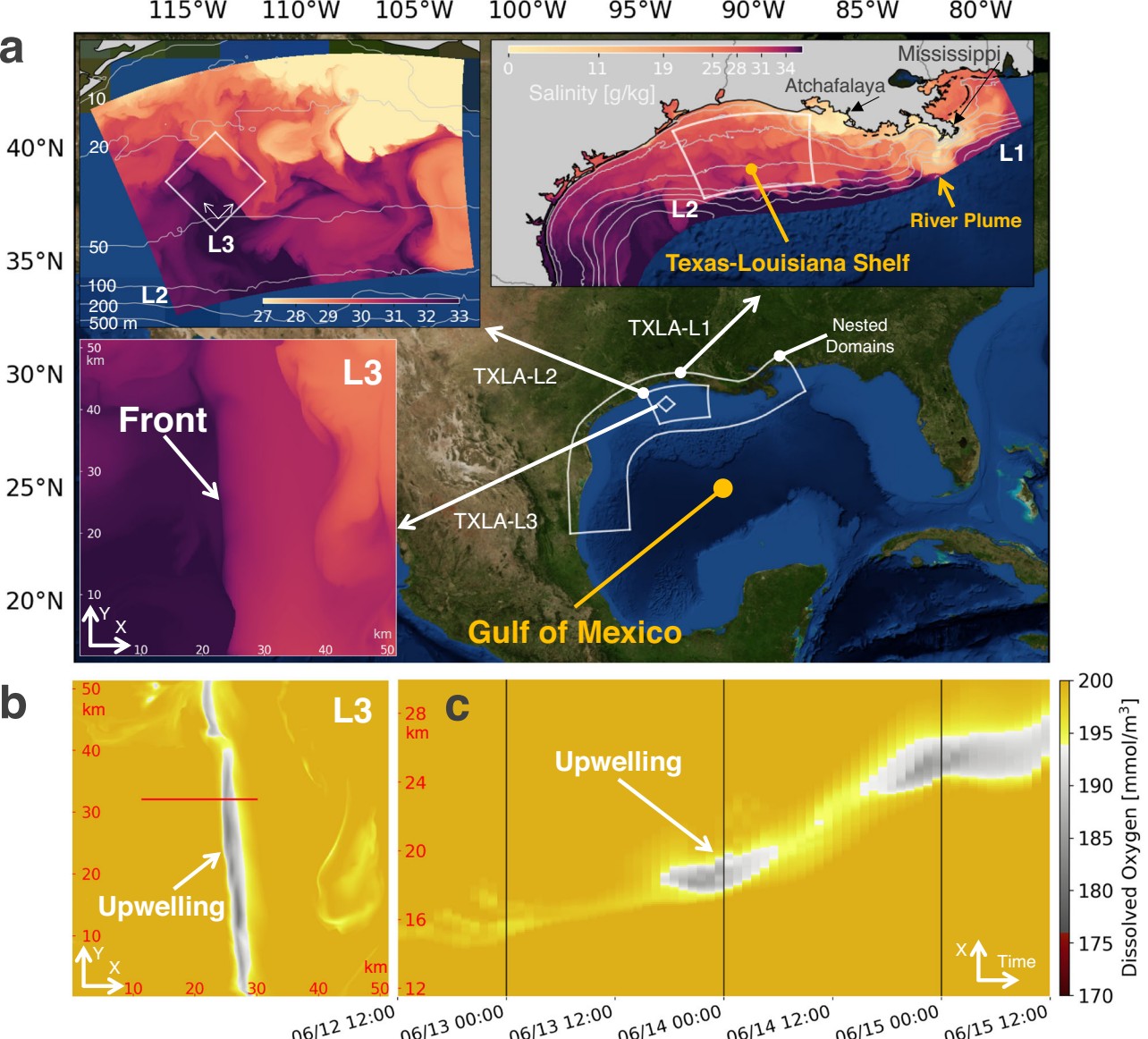

**Fig. 1 | Map of study site and triple-nested simulations. a** Northern Gulf of Mexico, Texas-Louisiana Shelf, and surface salinity from simulations. In summertime, the Mississippi and Atchafalaya rivers create a large river plume over the shelf with buoyant, relatively fresh water. Simulations termed TXLA (Texas-Louisiana) are triple-nested (see Methods for details). L1, L2, and L3 represent the nested layers. A front characterized by strong salinity gradients is highlighted in the L3 subpanel. Isobaths are contoured in gray in the L2 subpanel. **b** Horizontal slice of the dissolved oxygen at z = −4.7 m from the L3 simulation. Upwelling of water with lower oxygen is denoted. The upwelling is most prominent in L3 but less obvious in L1 and L2 due to the lower resolutions (Fig. S1 of the Supplementary Material). **c** Hovmöller diagram of dissolved oxygen. The diagram is made based on the oxygen field at z = −4.7 m along the section marked in **b**. The gray lines plotted every diurnal cycle indicate the diurnal upwelling of the water with lower oxygen. The wind forcing over the period is shown in Fig. 3d, showing a notable diurnal land-sea breeze. The snapshots in **a**, **b** are taken on June 15, 2010 00:00 UTC.

strong interactions between the inertial currents and the fronts on the shelf that could facilitate vertical exchange between the surface and bottom waters at the fronts. As part of a project funded by US NSF and UK NERC termed SUNRISE (Submesoscales Under Near-Resonant Inertial Shear Experiment), theoretical, modeling, and observational studies were undertaken to explore the vertical exchange induced by the interactions between the inertial currents and the fronts and how the oceanic fluid dynamics impact the marine environments over the shelf. A front-refined simulation under a condition with a notable land-sea breeze reveals upwelling of water with lower oxygen into the surface mixed layer at the front (Fig. 1b) that is modulated diurnally (Fig. 1c), suggestive of diurnal pulses of upwelling that draws bottom waters to the surface. High-resolution observational transects of a typical front on the shelf during a period of wind-forced inertial motions made in June

2021 also reveal evidence for slantwise vertical exchange at the fronts with streamers of high temperature surface waters (warmed by diurnal heating) descending to depth, and cooler waters upwelling on the fresher side (Fig. 2e, f). We will lead off the article describing these high-resolution observations of vertical exchange in further detail to motivate the study and then in subsequent sections use analyses of high-resolution simulations to understand the dynamics of the frontal vertical circulations that give rise to these motions and quantify their net impacts on the exchange of oxygen and other tracers on the shelf.

## Results

### SUNRISE campaign 2021
Characterizing the fronts on the Texas-Louisiana shelf and their vertical circulation is one of the goals of SUNRISE. In the summer of 2021,

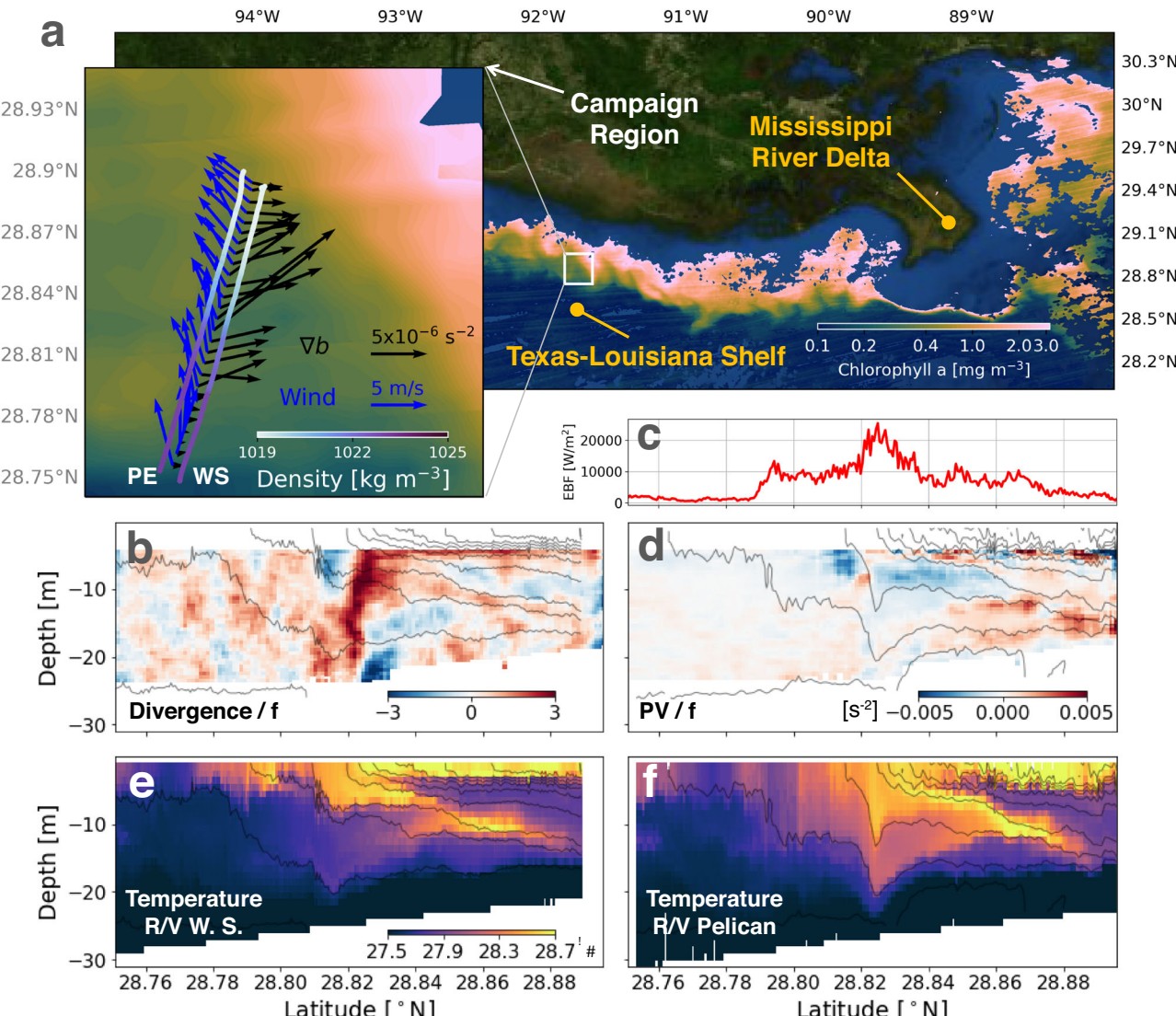

**Fig. 2 | SUNRISE Campaign 2021. a** Satellite imagery of surface chlorophyll a over the Texas-Louisiana shelf from NASA Suomi NPP VIIRS Ocean Color. The imagery is composited using the data collected between 18:00 and 20:00 UTC June 24, 2021. The subpanel is a zoomed in view of a region of the SUNRISE field campaign, where repeated (every 4 h), parallel transects were made by R/Vs Pelican (PE) and Walton Smith (WS) to sample the fronts in the Mississippi/Atchafalaya river plume. The two transects (roughly 17 km long each) were made between 03:00 and 07:00 UTC June 24, 2021 to sample the front which is characterized in the satellite imagery by strong gradients of chlorophyll. The front near-inertially oscillated by entering the sampling region and retreating. The ship tracks are colored by the near-surface density measured by VMPs. The blue arrows are the winds measured on R/V Walton Smith, and the black arrows are the horizontal buoyancy gradients ($\nabla_h b$) calculated using the near-surface density data from both R/Vs. **b** Section of normalized divergence ($\delta/f$), where $\delta = \partial u/\partial x + \partial v/\partial y$. **c** Profile of Ekman buoyancy flux $EBF = [\tau \times \hat{k}/(\rho_0 f)] \cdot \nabla_h b$, where $\tau$ is the wind stress and $\nabla_h b$ is the horizontal buoyancy gradient. The EBF is expressed in units of an effective heat flux, and positive values indicate a tendency for winds to destabilize the front. **d** Section of the potential vorticity (PV/f). **e, f** Temperature sections from the VMP profiling on the R/Vs. Isopycnals are contoured every 0.5 kg/m³ in gray.

a multi-platform field campaign was conducted on the Texas-Louisiana shelf. During the period from June 23 to 28, several wind events energized near-inertial oscillations in the ocean (Fig. S2 in the Supplementary Material). Over the shelf, the Mississippi/Atchafalaya river outflow creates a large plume of fresh water and a rich field of eddies and fronts develops as a result. A plume front was captured by high-frequency, repeated profiling transects conducted by R/Vs Pelican and Walton Smith which were run in parallel lines separated by ~1 km. A snapshot of the plume front (characterized by the high concentration of chlorophyll) and the ship tracks are shown in Fig. 2a. The observations revealed strong frontal convergence and divergence, the downward vertical transport of warm surface waters (through a process known as subduction) on the leading edge of the front, and upwelling of cool waters on its trailing edge.

Isopycnals near the surface converge approximately every inertial period, when the plume front propagates offshore. Here, we focus on the front observed on June 24. From 03:00 to 07:00 UTC, the plume front propagates into the sampling region, which is indicated by the strong surface buoyancy gradients at the middle of the transects (e.g., the subpanel in Fig. 2a). Strong near-surface convergence was found at the head of the front, suggesting intense downward motions beneath the front (Fig. 2b). As observed on both R/Vs, the isopycnals beneath the leading edge of the front are depressed due to downward advection of buoyant water, which is evidenced by the shock-like jumps in the density fields (Fig. 2e, f). In addition, strong near-surface convergence was found at the head of the front, suggesting intense downwelling beneath the front (Fig. 2b). The appearance of streamers of warm water on the leading edge of the front that follow isopycnals is strongly suggestive of

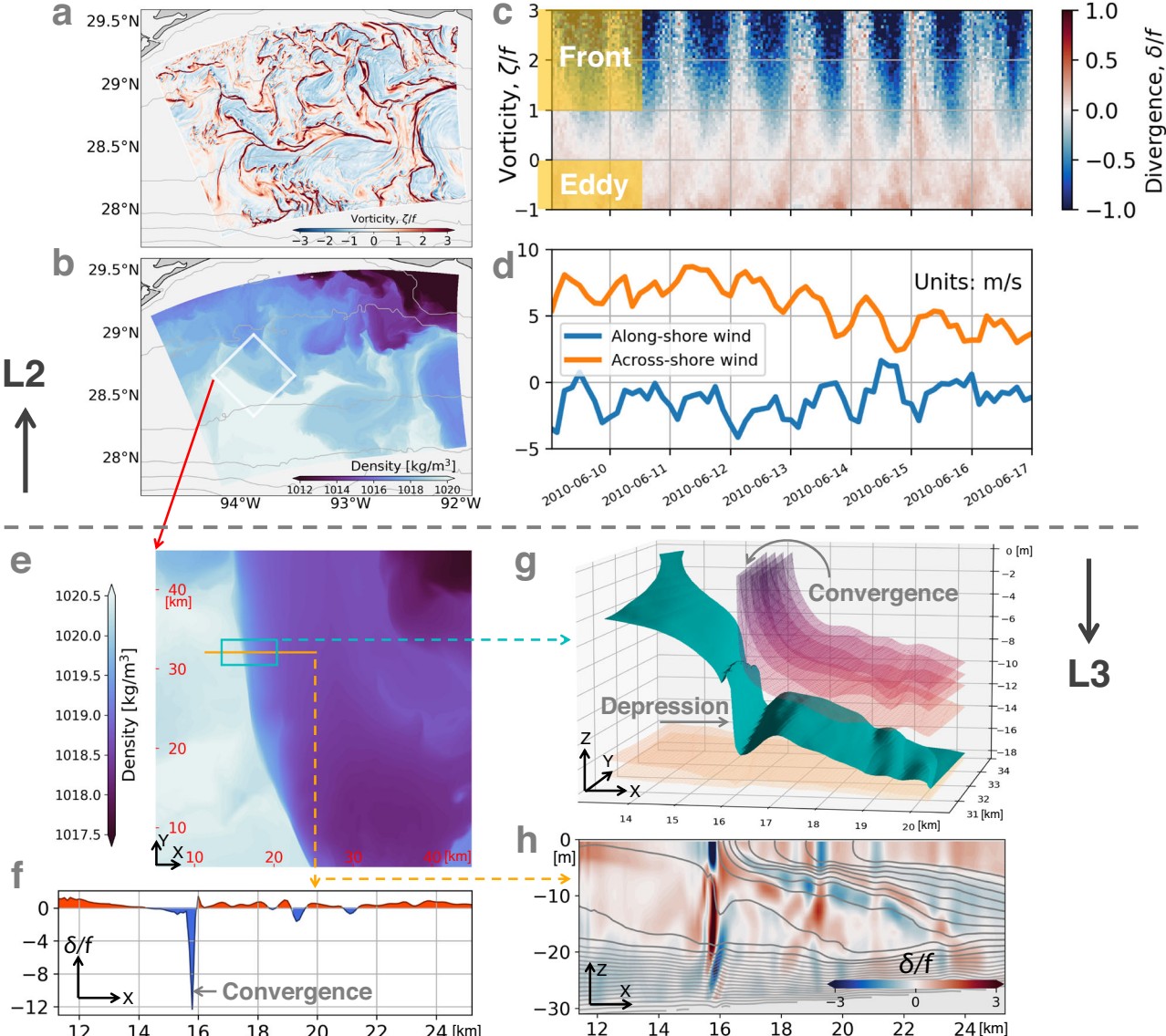

**Fig. 3 | Diurnal convergence at fronts. a** Surface normalized relative vorticity ($\zeta/f$), where $\zeta = \partial v/\partial x - \partial u/\partial y$. **b** Surface density. Isobaths are contoured in gray. The white box denotes the L3 domain. **c** Surface normalized divergence ($\delta/f$) binned as a function of $\zeta/f$ and time. The statistics are calculated over the L2 domain. **d** Time series of land-sea breeze. The wind speed is spatially averaged over the L2 domain. **e** Zoomed in view of surface density over the L3 domain. **f** Surface normalized

divergence ($\delta/f$) along the orange line marked in (**e**). Convergence (divergence) is shaded in blue (red). **g** 3D structure of isopycnal surfaces near the convergence zone (marked by the blue box in **e**). **h** Across-front section of $\delta/f$ along the orange line marked in (**e**). Isopycnal surfaces and contours are made every 0.2 kg/m³ in **g** and **h**. All the panels except **c** and **d** correspond to June 14, 2010 00:00 UTC. Panels **a**–**d** (**e**–**h**) are plotted from the L2 (L3) solution.

active subduction of surface waters at the front (Fig. 2e, f). In contrast, the near-surface flow is divergent on the light side of the front (Fig. 2b) which presumably drives the upward motions that occur there, as evidenced by the intrusions of cooler water (e.g., Fig. 2f). Such upwelling motions are also evidenced by the intrusion of bottom waters with lower oxygen and higher turbidity on the transects made on June 23 and 25 (see Fig. S3 and S4 of the supplementary material).

## Diurnal convergence at fronts

A triple-nested model is configured to simulate the frontal vertical circulations on the Texas-Louisiana shelf and understand their governing physics (Fig. 1a; see Methods for details). Over the shelf, a rich field of eddies and fronts develops due to the Mississippi/ Atchafalaya river outflow (Fig. 3a, b). In addition, the prominent wind variability in the summertime is often a diurnal land-sea breeze. An example of the land-sea breeze is shown in Fig. 3d. The

breeze resonantly excites strong near-inertial oscillations in the ocean (oscillatory flows where the Coriolis force acts as the sole restoring force for their motions). In the absence of background currents, inertial oscillations are non-divergent in the lateral direction. However, in the presence of the plume fronts, the oscillations interact with the fronts and display convergent features, leading to strong vertical motions at the front.

The plume fronts coincide with filaments with strong positive relative vorticity, while the plume eddies are characterized by negative relative vorticity (Fig. 3a, b). Given these characteristics, the horizontal divergence associated with the fronts and eddies can be distinguished from one another by calculating the divergence as a function of vorticity using surface fields taken from the entire numerical domain. The statistics of the divergence shown in Fig. 3c demonstrates that horizontal convergence (blue colors) primarily occurs in fronts and varies near-inertially with the sea-breeze.

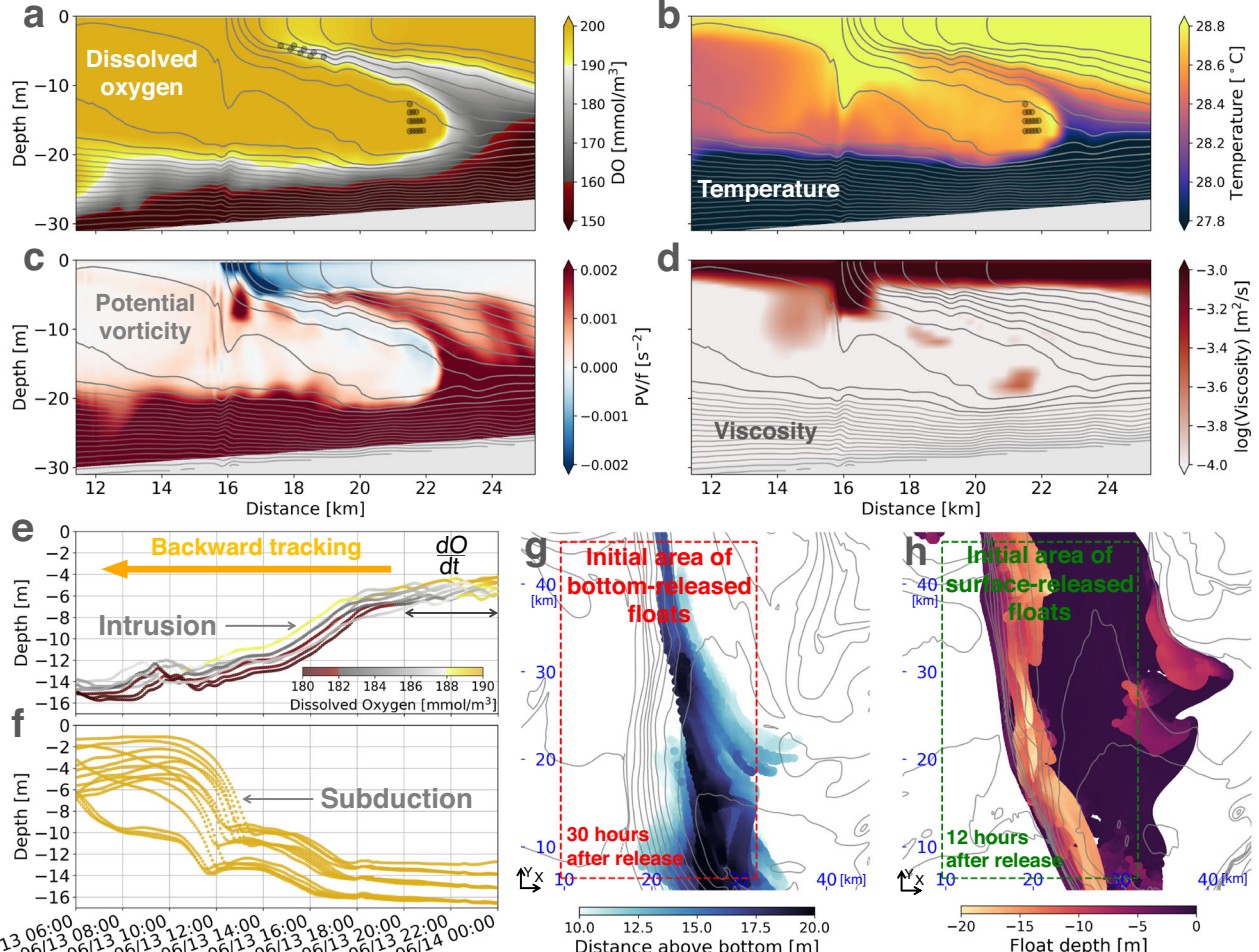

**Fig. 4 | Slantwise vertical exchange at fronts. a–d** Across-front sections of dissolved oxygen, temperature, potential vorticity, and viscosity. The sections are made at June 14, 2010 00:00 UTC along the orange line marked in Fig. 3e. **e** Time series of float depth colored by dissolved oxygen concentration. The floats are initially released at the positions marked by the upper gray dots in **a** and backward tracked in time. The Lagrangian rate of change of oxygen near the surface is estimated as $dO/dt \approx 0.82$ mg L$^{-1}$ day$^{-1}$. This rate of oxygenation is obtained by averaging the rates of all the floats over the last 4 h. **f** Same as **e** but for the lower floats marked in **a**, **b**. **g** Distance above bottom of floats 30 h after release. The floats are initially and uniformly released in a 10 m-thick bottom boundary layer within the red dashed box at June 13, 2010 06:00 UTC. **h** Float depth 12 h after release. The floats are initially and uniformly released at the surface within the green dashed box at June 13, 2010 12:00 UTC. Isopycnals are contoured every 0.2 kg/m$^3$ in gray in **a–d** and **g**, **h**. All the calculations are from the L3 solution.

The near-surface inertial/diurnal horizontal convergence at the fronts can induce downward motions at depth. The convergence at the surface across the front is found to be very strong, for example at $x \approx 16\ km$ for the section shown in Fig. 3f. This strongly convergent surface flow draws together isopycnal surfaces and induces downward motions which advect the buoyant surface water downwards and contributes to the depression of isopycnals beneath the leading edge of the front, a feature that is seen in the observations (cf. Figs. 2f and 3g). In addition to the convergence on the frontal edge, the flow at the light side of the front is divergent, resembling the observations (cf. Figs. 2b and 3h), and could potentially drive upward motions.

**Slantwise vertical exchange at fronts**
Evidence of subduction can be seen in the distribution of the temperature field near the fronts. The density of the sea water on the Texas-Louisiana self shelf is largely dependent on salinity with a lower-water column thermocline contributing to stratification differences in the summer[5]. Consequently, temperature becomes an ideal tracer to identify surface waters because it tags waters warmed by solar radiation. The simulation reveals a streamer of warm water subducting into the interior beneath the front, which resembles the observed fields (cf. Figs. 4b and 2e, f). A Lagrangian analysis was used to confirm that the

warm waters at depth originated from the surface. Namely, Lagrangian, or water-following floats were seeded in the simulation at the head of the warm-water streamer (the dots in Fig. 4b) and advected backwards in time to determine their origin. This Lagrangian analysis reveals that the floats came from near the surface and subducted a distance of about 10 m in a few hours, indicating rapid vertical transport (Fig. 4f). To illustrate the spatial structure of the subduction, floats were uniformly released at the surface of the frontal region (marked in Fig. 4h) and then tracked forward in time. As the floats approach the front, they are rapidly subducted beneath the front, showing a fairly uniform subduction zone along the front (Fig. 4h). The subduction has an along-isopycnal nature connecting the surface to the interior, which is different from other turbulent processes in surface mixed layers, such as breaking of surface gravity waves, convective instabilities, and shear instabilities.

Upwelling of bottom waters on the lighter side of the front accompanies the subduction of surface waters on the denser side of the front. This is particularly evident in the oxygen field. In the simulation, a bottom boundary layer with lower oxygen concentration compared to the overlying water is reproduced (details of the oxygen model are described in the Methods section), and hence oxygen is a particularly good tracer for identifying bottom waters (Fig. 4a). Interestingly, the

water with lower oxygen seems to intrude upward along the tilted isopycnals near the front (Fig. 4a). To certify the path of the intrusion, 3D Lagrangian floats are released at the tongue of the water with lower oxygen (denoted in Fig. 4a) and then tracked backward in time. The time series of the float depth shows that the floats at the tongue are raised about 10 m in 18 h (Fig. 4e), indicating that the near-surface water with lower oxygen certainly originates from a deeper depth. To illustrate the spatial structure of the intrusion, floats are uniformly released in a 10 m thick bottom boundary layer in the frontal region (marked in Fig. 4g) and then tracked forward in time. The floats are raised from the bottom boundary layer quite uniformly at the light side of the front (Fig. 4g), consistent with the intrusion of water with lower oxygen shown in Figs. 4a and 2b. Although the freshwater input from the Mississippi/Atchafalaya River generally suppresses ventilation of bottom waters by creating a strongly stratified layer, the exchange of water along isopycnals near the plume fronts serves as a ventilating channel of the bottom oxygen-deficient water over the Texas-Louisiana shelf that can bypass the plume's stratification barrier. This slantwise vertical exchange at fronts contributes a ventilation mechanism among other mechanisms related to bottom boundary layer dynamics[6].

A third tracer that evidences slantwise vertical exchange at the fronts is the potential vorticity (PV). The PV, defined here as $q = (f\hat{z} + \nabla \times \mathbf{u}) \cdot \nabla b$ ($\hat{z}$ is the unit vector in the vertical direction, $\mathbf{u}$ is the velocity, $b = -g\rho/\rho_o$ is the buoyancy, $g$ is the acceleration due to gravity and $\rho$, $\rho_o$ is the density and a reference density, respectively) is a measure of the stability of the flow. Namely, large positive values of the PV indicate stable flow and strong stratification, while negative values of the PV mark a flow that is unstable to submesoscale instabilities which can drive vertical exchange[7,8]. The structure of the PV field near the front looks remarkably similar to that of the temperature and oxygen fields (Fig. 4a–c), with high values of the PV (associated with the strongly-stratified bottom waters) being drawn up to the surface from the bottom in the plumes of waters with cooler temperature and lower oxygen. Conversely, water with low to negative PV is subducted from the surface ahead of the leading edge of the front along with the warmer, oxygenated waters. The PV inferred from the observations made from the two ships (see Methods for the calculation detail) shows a similar correlation between the PV and temperature fields (cf. Fig. 2d, f), suggesting that related PV dynamics are at play in the observed and simulated fronts.

The resemblances in the PV, temperature, and oxygen fields highlight the conservative nature of these three tracers. Having said this, the Lagrangian time evolution of the PV, as measured by the PV interpolated to the positions of the floats that were seeded into the bottom waters (colors in Fig. S5B of the supplementary material), show that the PV is conserved only until the water reaches the surface layer, at which point the PV is rapidly reduced and in fact changes sign. These floats upwell on the isopycnals that outcrop at the front, where the PV is negative in the surface boundary layer (Figs. 4c and 5b). Here nonconservative effects, namely friction and diabatic processes conspire to change the PV by inducing a positive vertical PV flux, $J_z$, at the sea surface that extracts PV from the ocean and hence reduces the PV in the boundary layer (e.g., Fig. 5c and Fig. S6 in the supplementary material where $J_z$ is defined and broken down into its frictional and diabatic components). At fronts forced by winds, the surface PV flux scales with the Ekman buoyancy flux, EBF $\equiv \mathbf{M}_e \cdot \nabla_h b$, where $\mathbf{M}_e$ is the Ekman transport, which measures the tendency for wind-forced surface flows to advect denser waters over light and destabilize the water column[8,9]. The EBF at the front oscillates in time due to both the variation in the wind-driven flow and the strength of the front (Fig. S7c in the supplementary material). Expressed in terms of an equivalent heat flux, the EBF can reach values approaching 20,000 W m$^{-2}$ of heat loss in the center of the front. Such intense values of the EBF were also inferred from the observations (Fig. 2c) and suggest that the surface boundary layer at the front is a hot spot for irreversible mixing.

The oxygen on the floats, like the PV, experiences similar rapid Lagrangian variations in the surface boundary layer, but in this case increasing as the fluid approaches the oxygenated surface ocean (Fig. 4e). In this way, by bringing bottom waters to the surface, the slantwise, along-isopycnal motions at the front can lead to irreversible changes in water properties, and drive a net oxygenation of the bottom waters and a reduction in the PV and stratification of the water column. The rate of oxygenation during these pulses of upwelling is estimated based on Lagrangian particle tracking as 0.82 mg L$^{-1}$ day$^{-1}$ (Fig. 4e), which is comparable to the rate of deoxygenation due to aerobic respiration of organic matter during the summertime in the Gulf of Mexico[10]. In the next section we explore the mechanisms that force the vertical motions at the front.

## Inertially modulated convergence and frontogenesis

The vertical motions at the front are ultimately associated with convergences and divergences in the near-surface horizontal flow. Understanding the processes that drive these convergences/divergences is thus key to unlocking the underlying physics of the vertical exchange at the fronts. As emphasized above, the convergence varies inertially and hence must be linked to the wind-forced inertial motions. However, in the absence of a background current, such motions inherit the horizontal scales of the winds, which on the shelf are too broad to explain the narrow convergence lines seen in the simulations, suggesting wave-mean flow interaction at the front as the cause of the convergence. By comparing the results from a reduced physics model that retains select terms in the equations of motion to the full solution from the simulations, we were able to isolate the key wave-mean flow interaction that is responsible for the inertially-modulated convergence at the fronts.

The reduced physics model accounts for wind-forcing, the Coriolis force, the pressure gradient force at the front, mixing of momentum by turbulence, and accelerations, but not nonlinear advection of momentum in the equations of motion (namely, it is a variant of the so-called turbulent thermal wind balance, e.g., ref. 11, that allows for accelerations but it does not account for modifications of the inertial motions by vorticity; see the Methods section for the details of the reduced physics model). In spite of the model simplification, the model captures the features in the cross-front velocity that lead to the inertially-modulated convergence (cf. the two panels in Fig. 5g). Here, we highlight a period of strong convergence around 6/14 18:00 associated with the sharp reversal in the cross-front velocity within the front relative to outside of the front (e.g., Fig. 5e, g for cross-front distances less than 3 km and before 6/14 21:00). The elevated turbulent viscosity at the front indicates that the mixing of the geostrophic momentum could play an important role in driving ageostrophic frontal circulations (Fig. 4d). The reduced physics model reveals that the flow reversal is caused by mixing of the geostrophic flow, which disrupts the geostrophic balance and accelerates an ageostrophic flow $u_{ag} < 0$ that streams water down the pressure gradient at the front. Mixing of geostrophic momentum is quantified by the "geostrophic stress", $\boldsymbol{\tau}_{\mathbf{g}} = -\rho_o \nu \hat{z} \times \nabla_h b / f$ ($\nu$ is the turbulent viscosity)[12]. A time series of the geostrophic stress evaluated on the light side of the front demonstrates how reversals in the ageostrophic flow follow the peaks in $|\boldsymbol{\tau}_{\mathbf{g}}|$ (cf. Fig. 5d, g) which in turn coincide with the maxima in the turbulent viscosity (e.g., Fig. S7b in the supplementary material). These maxima in $\nu$ occur during periods when the EBF is positive and wind-driven surface flows destabilize the water column and enhance mixing (e.g., Fig. S7c in the supplementary material).

On the dense side of the front the winds drive an oscillatory, ageostrophic flow that tracks the phasing of a slab mixed layer model (an even simpler reduced physics model that only accounts for wind-forcing, the Coriolis force, and accelerations; see the Methods section for the details), cf. the two panels in Fig. 5f. The ageostrophic flows on

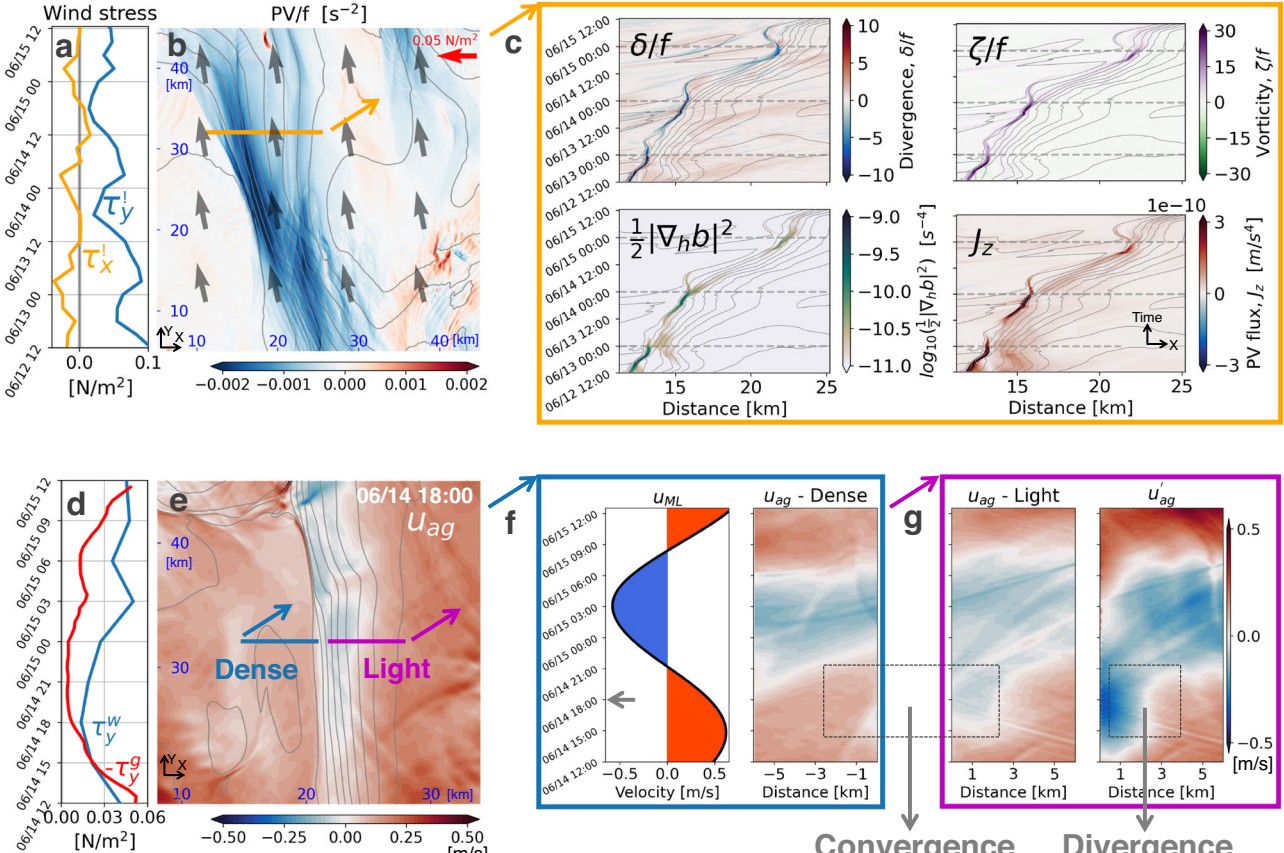

**Fig. 5 | Inertially modulated frontogenesis and convergence. a** Time series of the surface wind stress over the L3 domain. **b** Surface PV (color), wind stress (vectors) and density (contours) that are averaged between 06/12 12:00 and 06/15 12:00 UTC. **c** Hovmöller diagrams of surface normalized divergence ($\delta/f$), horizontal buoyancy gradient squared ($1/2|\nabla_h b|^2$), and PV flux ($J_z$), which are calculated along the orange line in **b** (the same line in Fig. 3e). **d** Comparison between the wind stress $\tau_y^w$ and minus the geostrophic stress $\tau_y^g$ at the front over one inertial period. $\tau_y^g$ and $\tau_y^w$ are averaged over a front-following section (the first 2 km of the magenta section in **e**). **e** Across-front ageostrophic velocity $u_{ag}$. The velocity is averaged over the top 5 m. The snapshot is taken at 06/14 18:00 UTC (marked by the arrow in **f**). Two front-following sections are plotted for each side of the front. **f** Velocity evolution over one inertial period on the dense side of the front. The left subpanel shows one inertial cycle of $u_{ml}$ that is obtained from the slab mixed layer model. The right subpanel is a Hovmöller diagram of $u_{ag}$ along the blue line in **e**. **g** Velocity evolution over one inertial period on the light side of the front. The left subpanel is a Hovmöller diagram of $u_{ag}$ along the magenta line in **e**. The right subpanel is a similar diagram but for $u'_{ag}$ that is obtained from the reduced physics model. The gray dashed box between **f** and **g** marks the phase when the across-front velocities run opposite on either side of the front and lead to convergence. The gray box in **g** marks the phase with divergence on the light side of the front. The distance in the Hovmöller diagrams in **f** and **g** are front-relative with negative (positive) values on the dense (light) side. The gray contours in **b**, **c**, **e** are isopycnals made every 0.2 kg/m³. All calculations are made from the L3 solution.

either side of the front thus follow different physics and have periods when they are opposing (e.g., between 15:00–21:00 on 6/14). This results in strong, diurnal convergence ($\delta/f < -5$) at the front (Fig. 5c). The near-surface convergence leads to frontogenesis as it intensifies the lateral buoyancy gradient, $\nabla_h b$, and cyclonic vorticity ($\zeta > 0$) of the front, which is likewise modulated over an inertial period (Fig. 5c). Based on this analysis we conclude that this inertial modulation of the convergence and frontogenesis is ultimately attributable to periodic mixing of the geostrophic flow at the front caused by the diurnally-varying winds. Similar diurnal variations in frontogenesis have been modeled for fronts forced by diurnal heating and cooling which can induce variations in viscosity akin to the wind-driven modulations in turbulence emphasized here[13,14].

The reversal of the ageostrophic flow at the front also leads to divergence on the light edge of the front (e.g., Fig. 5c, g between 2–3 km). The pattern of near-surface convergence and divergence on the dense and light sides of the front should set up a vertical circulation with downwelling and upwelling, respectively. Furthermore, theory predicts that overturning motions forced by near-surface convergence/divergence at fronts should be slantwise, with streamlines that tend to run parallel to isopycnals[15,16], which is

entirely consistent with the behavior of the tracer fields and Lagrangian floats seen in the simulation. A schematic highlighting the key elements contributing to the vertical circulation at the front is shown in Fig. 6.

## Discussion
Buoyant material (including microplastics, oil droplets and phytoplankton) aggregate at fronts due to convergent surface currents[17–20]. Aggregation of *Sargassum* is well known to occur at the boundaries between water masses, and was observed at the fronts of the river plume along with strong convergence during SUNRISE Campaign 2021. The downward motions associated with the diurnal convergence might counteract the upward motion of buoyant material, making it possible that some of the buoyant material could be subducted beneath the front.

The fronts studied here develop in regions with significant sub-mesoscale eddy activity (see Figs. 2a and 3a, b). The magnitude of the modeled subduction associated with the frontal convergence is similar to that previously seen at the leading edge of a primary river plume[21–24], where the density differences can be significantly larger than the 1–2 kg/m³ seen here. Since many fronts, similar to the ones studied

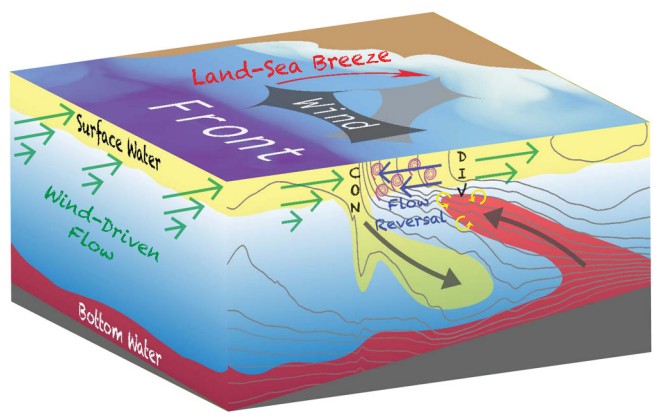

**Fig. 6 | Schematic of the key processes contributing to the vertical circulation.** A diurnal land-sea breeze generates an inertially-oscillating wind-driven surface flow. The wind-driven flow interacts with an ocean front and transports dense water over light water at certain phases over an inertial cycle, destabilizing the water column and enhancing turbulence and mixing at the front. The mixing of the geostrophic flow by the turbulence creates a counterflow at the front, which runs opposite to the wind-driven flow and leads to strong convergence (abbreviated as CON). Similarly, the flow reversal also leads to divergence (abbreviated as DIV) at the light side of the front. The near-surface convergence and divergence at the front set up a slantwise circulation along isopycnals in the interior, inducing downwelling of surface water and upwelling of bottom water. This vertical exchange can insert lower dissolved oxygen waters into the surface mixed layer, where oxygenation can be stimulated by mixing processes (schematized by the yellow curly arrows).

here, develop at the edge of eddies (see Fig. 3a, b), strong subduction at such fronts could be a ubiquitous process across the shelf during periods of strong eddy activity.

Upwelling within the front of bottom hypoxic waters over the Northern Gulf of Mexico shelf has been observed in numerical models[6,25], but the vertical motions seen here are more energetic, indicating an even greater ventilation of the bottom boundary layer as well as interaction with surface waters. Our numerical simulation shows that in frontal zones the near-surface, oxygenated water is subducted into the bottom boundary layer in ~8 h, and the near-bottom water with lower oxygen intrudes into the surface mixed layer in ~12 h (Fig. 4e, f). In addition, the simulation suggests that the upwelled water can be quickly oxygenated when the water mass approaches the surface mixed layer. The vertical motions associated with the frontal processes described here could serve as a ventilation mechanism for the bottom waters on the shelf and play a potential role in the dynamics of dead zones in the region. Additionally, the introduction of nutrient-rich water into the photic zone may play a potential role in recycling nutrients over the shelf.

The slantwise vertical exchange with displacements of $O(10\ m)$ over a 8–12 h period could be compared to turbulent mixing through scaling $\kappa \sim L^2/T \sim$ 2.3 to $3.4 \times 10^{-3}$ m$^2$ s$^{-1}$. This is large compared to typical values for small-scale turbulence in the stratified interior (e.g., the interior mixing from the simulation is one order of magnitude smaller as shown in Fig. 4d). A critical difference here is that the slantwise motions do not initiate a turbulent cascade at the overturn scale, so the water is not homogenized through further mixing, yet the primary exchange is not easily reversible since the slantwise overturning motions in the interior are driven by irreversible mixing processes in the surface boundary layer. Since tides are relatively weak in the Northern Gulf of Mexico[4], and energetic wind forcing due to atmospheric frontal passages is also infrequent[3], it is likely that the vertical motions driven by land-sea breeze described here are a dominant vertical exchange mechanism across the main pycnocline during summer.

## Methods
### Observational platform
The SUNRISE Campaign was launched in the summer of 2021 from June 19th to July 9th. R/Vs Walton Smith and Pelican were employed to conduct the field campaign. The data collected during the campaign include hydrography (such as temperature and salinity) from Vertical Microstructure Profiler (VMP) casts, and velocity measurements made using 600 kHz (pole-mounted) and 1200 kHz (ship-mounted) Acoustic Doppler Current Profilers (ADCPs). Two R/Vs were coordinated to simultaneously sample on parallel transects separated by 1 km. Repeated adaptive transects were made to follow the river plume fronts. The transects were approximately 20 km long, each taking nearly 4 h. The data of the VMP casts were processed at a vertical resolution of 1 m, with average spacing between casts of 110 m. The data of the 600 and 1200 kHz ADCPs were averaged over 2 min and processed at vertical resolutions of 1 m and 0.5 m, respectively, using UHDAS + CODAS. Utilizing the synchronized observations from the R/Vs, the buoyancy gradients, divergence, relative vorticity, and potential vorticity were calculated using the plane-fitting method[26], without the mixing of spatial and temporal aliasing.

### Numerical simulations
The numerical simulations termed TXLA (Texas-Louisiana) are configured in the Regional Ocean Modeling System (ROMS) and the Coastal and Regional Ocean COmmunity model (CROCO)[27,28]. The simulations are based on a different year (2010) than the field campaign but under similar wind conditions. The model domains are selected for the region of rich submesoscale fronts over the Texas-Louisiana shelf. There are three nesting layers - TXLA-L1, L2, and L3. The L1 domain covers the entire shelf and its slopes with the horizontal resolution varying between approximately 650 m to 3.7 km. The L2 domain covers the central part of the shelf with a horizontal resolution of roughly 300 m. The L3 domain focuses on a specific submesoscale front with a horizontal resolution of 100 m. The bathymetry of the L2 and L3 domains is obtained by linearly interpolating the bathymetry of the L1 domain. The vertical grid of each domain has 30 terrain-following layers and has the same S-coordinate stretching parameters. The layers are concentrated near the surface and bottom to resolve the processes in the surface and bottom boundary layers. The L1 model is nested into Global HYCOM Reanalysis, the L2 model is online, two-way nested into L1, and the L3 model is offline, one-way nested into L2. L1 and L2 employ the hydrostatic ROMS, and L3 employs the CROCO-NBQ that is built upon CROCO with a non-hydrostatic kernel. Surface forcing and fluxes are obtained from ERA interim[29]. The data of river discharge are obtained from U.S. Army Corps of Engineers and U.S. Geological Survey. $k - \epsilon$, a generic length-scale turbulence closure model is used to parameterize the vertical mixing[30]. All the three models provide velocity, temperature, salinity, and dissolved oxygen. The L1 simulation was run for the period from 1994 to 2016. The L2 simulation was run for the period from June 1 to July 26, 2010 (roughly 6 weeks). The L3 simulation was run for the period from June 10 to 16 when the wind condition resembles the scenario of the field campaign. The parent model L1 has been extensively compared against observations to understand the model's strengths and weaknesses[31,32]. A separate study demonstrated that the model was statistically able to replicate the magnitude and characteristics of the eddies[5]. The model has been used in a variety of dynamical[33–35], environmental[25,36,37], and ecological[38–40] studies.

### Oxygen model
The dissolved oxygen is simulated as a tracer in the triple-nested simulations. The riverine water is saturated in oxygen. The oxygen at the lateral open boundaries is relaxed to the parent models. The gas exchange at the sea surface is assumed to be instantaneous, and the surface values of oxygen are set to saturated values based on

temperature and salinity[41]. The effects of photosynthesis on the concentration of dissolved oxygen are not included. A benthic respiration formulation, as the sink of oxygen, is applied at the bottom to provide the bottom oxygen flux following the respiration parameterization[42].

### Reduced physics model

A reduced physics model is used to illustrate how the front influences the wind-driven inertial oscillations. The model is linear and constructed as

$$\frac{\partial u}{\partial t} - fv = \overline{-\frac{1}{\rho}\frac{\partial p}{\partial x}} + \overline{\frac{\partial}{\partial z}\left(\nu\frac{\partial u}{\partial z}\right)}, \tag{1}$$

$$\frac{\partial v}{\partial t} + fu = \overline{-\frac{1}{\rho}\frac{\partial p}{\partial y}} + \overline{\frac{\partial}{\partial z}\left(\nu\frac{\partial v}{\partial z}\right)}, \tag{2}$$

where $(u, v)$ is the mixed layer velocity, $f$ is the Coriolis parameter, $(-\frac{1}{\rho}\frac{\partial p}{\partial x}, -\frac{1}{\rho}\frac{\partial p}{\partial y})$ is the pressure gradient force vertically averaged in the mixed layer, $(\frac{\partial}{\partial z}(\nu\frac{\partial u}{\partial z}), \frac{\partial}{\partial z}(\nu\frac{\partial v}{\partial z}))$ is the frictional body force vertically averaged in the mixed layer, and $\nu$ is the turbulent viscosity. The reduced physics model is forced by the terms on the right-hand side of the equations and solves for $(u, v)$. The reduced physics model is initialized with the velocity from TXLA-L3 along a section across the front (i.e., the magenta section in Fig. 5e) at June 12, 2010 12:00 UTC. The model outputs the mixed layer velocity as a function of across-front distance (in the x direction) and time. The pressure gradient force and frictional body force are extracted along the across-front section using the diagnostics of the momentum equation from the TXLA-L3 solution, vertically averaged within the top 5 m, and implemented as the forcing terms of the reduced physics model. The model is integrated for 3 days to get $(u, v)$, and then the ageostrophic velocity $(u_{ag}, v_{ag})$ is calculated by subtracting the geostrophic velocity (calculated using the vertically averaged pressure gradient force) from $(u, v)$. The evolution of the ageostrophic velocity $u_{ag}$ at the light side of the front during the last inertial cycle is shown in the right panel of Fig. 5g. $u_{ag}$ is denoted with a prime to distinguish from the ageostrophic velocity from the L3 simulation.

The frictional forcing term in the reduced physics model can be interpreted in a form with stresses. The vertically averaged frictional force can be expressed as

$$\overline{\frac{\partial}{\partial z}\left(\nu\frac{\partial \mathbf{u}}{\partial z}\right)} = \frac{1}{H_{ML}}\int_{-H_{ML}}^{0}\frac{\partial}{\partial z}\left(\nu\frac{\partial \mathbf{u}}{\partial z}\right)dz = \frac{1}{H_{ML}}\left[\left(\nu\frac{\partial \mathbf{u}}{\partial z}\right)\Big|_{z=0} - \left(\nu\frac{\partial \mathbf{u}}{\partial z}\right)\Big|_{z=-H_{ML}}\right], \tag{3}$$

where $H_{ML}$ is the thickness of the surface mixed layer. At the surface, the shear stress is equal to the wind stress,

$$\left(\nu\frac{\partial \mathbf{u}}{\partial z}\right)\Big|_{z=0} = \frac{\tau^{\mathbf{w}}}{\rho}. \tag{4}$$

At the bottom of the mixed layer, the shear has a component associated with the thermal wind shear $\frac{\partial \mathbf{u_g}}{\partial z}$, yielding a geostrophic stress $\tau^{\mathbf{g}} \equiv -\rho\nu\frac{\partial \mathbf{u_g}}{\partial z}$. Similarly, the ageostrophic shear $\frac{\partial \mathbf{u_{ag}}}{\partial z}$ is corresponding to an ageostrophic stress $\tau^{\mathbf{ag}} \equiv -\rho\nu\frac{\partial \mathbf{u_{ag}}}{\partial z}$. With these definitions, the shear stress at the bottom is

$$\left(\nu\frac{\partial \mathbf{u}}{\partial z}\right)\Big|_{z=-H_{ML}} = \left(\nu\frac{\partial \mathbf{u_g}}{\partial z}\right)\Big|_{z=-H_{ML}} + \left(\nu\frac{\partial \mathbf{u_{ag}}}{\partial z}\right)\Big|_{z=-H_{ML}} = -\frac{\tau^{\mathbf{g}}}{\rho} - \frac{\tau^{\mathbf{ag}}}{\rho}. \tag{5}$$

Consequently, the vertically averaged frictional force can be expressed as

$$\overline{\frac{\partial}{\partial z}\left(\nu\frac{\partial \mathbf{u}}{\partial z}\right)} = \frac{1}{H_{ML}}\left(\frac{\tau^{\mathbf{w}}}{\rho} + \frac{\tau^{\mathbf{g}}}{\rho} + \frac{\tau^{\mathbf{ag}}}{\rho}\right). \tag{6}$$

### Slab mixed layer model

A slab mixed layer model is used to calculate the inertial oscillations generated by winds in the surface mixed layer. The inertial oscillations shown in the left panel of Fig. 5f are calculated using the model:

$$\frac{\partial u_{ML}}{\partial t} - fv_{ML} = \frac{\tau_x}{\rho_0 H_{ML}}, \tag{7}$$

$$\frac{\partial v_{ML}}{\partial t} + fu_{ML} = \frac{\tau_y}{\rho_0 H_{ML}}, \tag{8}$$

where $(u_{ML}, v_{ML})$ is the mixed layer velocity, $f$ is the Coriolis parameter, $(\tau_x, \tau_y)$ is the surface wind stress, $\rho_0 = 1025$ kg/m$^3$ is the reference density, and $H_{ML} = 5$ m is the mixed layer thickness. The model is initialized with the ageostrophic velocity from TXLA-L3 at June 12, 2010 12:00 UTC, which is obtained by subtracting the geostrophic velocity from the total velocity and then spatially averaged over a section on the dense side of the front (i.e., the blue section in Fig. 5e) and within the top 5 m. The wind stress is the same stress used to force TXLA-L3 but spatially averaged over the frontal region. The model is integrated for 3 days to get a time series of $(u_{ML}, v_{ML})$, and $u_{ML}$ during the last inertial cycle is shown in the left panel of Fig. 5f.

### Reporting summary

Further information on research design is available in the Nature Research Reporting Summary linked to this article.

## Data availability

The satellite imageries used in this study are from the Suomi-NPP/VIIRS Ocean Color Data Product and available online through the WORLDVIEW portal supported by NASA: https://worldview.earthdata.nasa.gov. The map background is the NASA Blue Marble which is available at https://visibleearth.nasa.gov/collection/1484/blue-marble. The bathymetry dataset used in the TXLA simulation is ETOPO5 which is available at https://www.ngdc.noaa.gov/mgg/global/etopo5.HTML. The simulation forcing datasets, Global HYCOM Reanalysis and ERA-interim, are available at http://www.hycom.org and https://www.ecmwf.int/en/forecasts/datasets/reanalysis-datasets/era-interim, respectively. The river discharge data are obtained from U.S. Army Corps of Engineers (https://www.mvn.usace.army.mil) and U.S. Geological Survey (https://waterdata.usgs.gov/nwis). The data of the TXLA simulation and the forward/backward particle tracking are available online at https://doi.org/10.5281/zenodo.6381139. The data of the SUNRISE Campaign 2021 used in this study are available online at https://doi.org/10.5281/zenodo.6381027.

## Code availability

The source code of ROMS model is available at https://www.myroms.org. The source code of CROCO model is available at https://www.croco-ocean.org. The source code of the ADCP processing software, UHDAS+CODAS, is available at http://uhdas.org. The cartographic package used for the visualization in this study is Cartopy, and its source code and the associated geospatial data are available at https://github.com/SciTools/cartopy. The code used for the particle tracking is Pyticles, which is designed by Jonathan Gula and archived at https://doi.org/10.5281/zenodo.4973786. The code used for the post-processing of simulation output is XROMS[43], which is designed by Robert Hetland and archived at https://github.com/hetland/xroms.

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

## Acknowledgements

We thank the Master and crew of the R/V Pelican and the Master and crew of the R/V Walton Smith for their immense capability and tireless efforts that made these observations possible. All computing and data storage resources were provided by Stanford High Performance Computing Center and Texas A&M High Performance Research Computing.

This work was supported by the NSF (OCE-1851450 for LNT and LQ, OCE-1851531 for RKS and JDN, OCE-1851201 for JAM, and OCE-1851470 for DK and RDH) and NERC: NE/T004223/1. This work was partially supported by the Earth System Model Development and Regional and Global Modeling and Analysis program areas of the U.S. Department of Energy, Office of Science, Office of Biological and Environmental Research as part of the multi-program, collaborative Integrated Coastal Modeling (ICoM) project.

## Author contributions

Conceptualization: L.N.T., A.F.W., L.Q. Investigation: L.Q., L.N.T., A.F.W., R.D.H., D.K., J.R.T., F.H.W.H., R.K.S., J.D.N., J.A.M. Visualization: L.Q., L.N.T., A.F.W., R.D.H., F.H.W.H., J.A.M. Writing–original draft: L.Q., L.N.T., R.D.H. Writing–review & editing: L.N.T., R.D.H., A.F.W., D.K., J.R.T., R.K.S., J.D.N., J.A.M.

## Competing interests

The authors declare no competing interests.
