## [Peer Review File · Nature Communications]

Rapid vertical exchange at fronts in the northern Gulf of MexicoEditorial Note: Parts of this Peer Review File have been redacted as indicated to remove third-party material where no permission to publish could be obtained.

REVIEWER COMMENTS

Reviewer #1 (Remarks to the Author):

What are the noteworthy results? The physics of frontal processes.

Will the work be of significance to the field and related fields? yes, if the modulation of the importance of processes relevant to low dissolved oxygen in the area of the measurements and the broader nGoMex continental shelf.

Does the work support the conclusions and claims, or is additional evidence needed? The physics is supported, but the extrapolations to shelf hypoxia are not.

Are there any flaws in the data analysis, interpretation and conclusions? The conclusions just need to be presented within the context of the data.

Is the methodology sound? Yes, and innovative. The extrapolations to continental shelf low oxygen is not supported.

Is there enough detail provided in the methods for the work to be reproduced? Yes, with appropriate funding.

Comments

Rapid Vertical Exchange at Fronts in the Northern Gulf of Mexico, manuscript comments

The experimental design for the physical oceanography observations is elegant with the simultaneous coordinated measurements of two ship transects across a well-defined front. The physics identified along the frontal edge are thorough and groundbreaking. And, there was an obvious intrusion of bottom-water along the convergence towards the surface, and vice-versa, as identified from dissolved oxygen concentrations overlaid on the physics. The numerical simulations and explanations are well done and succinct. It should be emphasized in the text where simulations versus observations are presented.

More informal language, e.g., "indeed," "so," "breathe," even if given single quotes, is up to the editors.

Study area figure 1: need to identify the Atchafalaya delta so that the reader (uninformed of nGoMx geography) understands the idea of the Mississippi/Atchafalaya plume, which is then dropped to Mississippi River plume, possibly describe the two river plumes, relative inputs, general downcoast flow in June, in Figure 1 and associated text.

"These winds generate strong inertial oscillations, which set up a diurnally-pulsing vertical circulation at the fronts that draws bottom waters up to the surface mixed 21 layer." This mechanism is verified by the dissolved oxygen concentrations."

AND

"Abstract: The simulations suggest that during these "breaths" the rate of oxygenation of the bottom waters is comparable to deoxygenation by the respiration of organic matter over the shelf and hence could play an important role in the evolution of the region's dead zone.

Comment:

L130, "near-surface low-oxygen water." It would be helpful if the range of concentrations could be provided in the text. However, I can surmise that none of the values qualify for being "hypoxic," given a range of 150-200 μM , which converts to approximately 80 to 100% saturation. This range of dissolved oxygen is in the upper range of values expected in this area in June, not near "hypoxic," and the bottom-waters do not approach "hypoxia."

Figure 3H, indicates that the horizontal width of a 'bore' that reaches ~ 30 m is less than 1 km and similar but not as well-defined to depths of 10-15 m.

Given the above results related to dissolved oxygen, it seems the intrusion/extrusion of different levels of dissolved oxygen is a minor process in the overall dissolved oxygen fields on this area of the Louisiana shelf. The last sentence of the abstract should be revised away from "an important role"; the data do not support the existing statement.

The diurnal cycles are the focus but there is no mention of the tidal cycles and their interactions, which should be present at this location. If they were part of the analysis, this needs to be stated,

especially, if they were not identified in the physical environment. And, if not examined, this should also be stated.

Specifics:

L14, suggest "support hypoxia formation and maintenance" instead of "condition"

18, "...exchange at these fronts can be quite rapid and can lead to the oxygenation of bottom water" This reviewer will look for frontal vertical mixing within the depth gradient where hypoxia is likely to occur. Much of the plume front is over much deeper continental shelf areas than where hypoxia occurs.

21, these "breaths" the rate of oxygenation of the bottom waters is comparable to deoxygenation by the respiration of organic matter over the shelf. This reviewer will look for temporal respiration rates typical of shelf ecosystems comparable to those generated to the "breaths" observed by the authors.

33-34, suggest connecting the surface waters with the bottom waters.

L34-35, can evade the stratification barrier of the plume; meaning not clear

L 61, define "SUNRISE" here

L 70, better description of Bore-like features

L73 more technical description of shock-like jumps

L76, ADCP observations that substantiate the down-welling?

L79, suggest "cooler" water; the temperature range is 28.76 to 28.88 °C in Figure 2 and 27.8 to 28.8 in Figure 4.

L102, suggest something other than "nose." Possible, peak, apex...

L108, is largely dependent on salinity for most of the year [addition: with a lower-water column thermocline contributing to stratification differences in the summer].

L115,... indicating rapid vertical transport... However, as noted in other comments, these intrusions are of limited geographical expanse and ephemeral.

L123,... particularly evident in the oxygen field... As noted in other comments the dissolved oxygen range is minimal and not "hypoxic."

L208, suggest "denser"

L229, should italicize Sargassum

L251-252, statements such as "...could serve as a significant ventilation mechanism for the bottom waters on the shelf and play an important role in the evolution of dead zones in the region." are not supported by the data or the simulations. The data in this manuscript represent a period in early June, and the low-oxygen area persists substantially from June – August or September, under strengthened stratification and long periods of no re-aeration. Suggest "dynamics" instead of "evolution," and "potential" for "important role."

L252, ... introduction of low-oxygen, nutrient-rich water into the photic zone may play an important role in..." agree with the concept, but it needs proof. Suggested wording "introduction of nutrient-rich water associated with low oxygen..." It is not the low oxygen that is important in the process but the nutrient-rich bottom waters.

L254, ... Frontal processes create strong temporal variability... As noted the temporal duration is minimal, and the dissolved oxygen differences are also minimal. Suggest that this finding be reworded.

L255, suggest that the wording should be "...to be an environment in which hypoxia..."

L256, suggestremains unclear...

L263, the idea of tidal processes is just now being introduced. Suggestions made earlier about tidal cycles. Tidal advection is more prominent where the depth gradient is steeper, not in the small gradient of the study area.

L266, Information about the frequency of frontal passages in the summer in the nGoMex are available in the literature and should be included.

L275, delete "fulfill."

L300, the surface waters may be super-saturated in dissolved oxygen at this time of year, despite the warm water temperatures.

References: please make uniform according to the journal format, e.g., L354, capitalize journal name, here and elsewhere; L356, insert ..., USA after PNAS;

L429, should the URL link be provided here?

Excellent graphics, but clarity on some could be improved.

Figure 3. Can a more technical word than "bore" be used, e.g., "intrusion"?

The infographic (Fig. 6) is well done and informative. I suggest an arrow for North be included.

Figure 6 caption: Schematic of the key processes contributing to the vertical circulation. A diurnal land-sea breeze wind-driven generates [and] inertially-oscillating surface flow. Should the "and" be "an"?

Figure 6 caption, ... a bottom oxygen-deficient water mass can "breathe" through a channel reaching the surface mixed layer, where oxygenation can be stimulated by various physical and biochemical processes.

Suggest,can insert lower dissolved oxygen waters into the surface mixed layer through....

Deletestimulated by various physical [delete and biochemical] processes Biogeochemistry is not included in this manuscript.

Review of **Rapid Vertical Exchange at Fronts in the Northern Gulf of Mexico**

In the article *Rapid Vertical Exchange at Fronts in the Northern Gulf of Mexico* by Lixin Qu et al, the authors investigate the vertical motions happening in frontal regions forced by the land-sea breeze over the Northern Gulf of Mexico and highlight potentially large impacts for oxygenation of bottom waters.

The study uses a unique set of observations, state-of-the-art numerical simulations, and elaborated diagnostics to characterize vertical motions happening at fronts and the mechanisms driving them. Observations come from a dedicated cruise (SUNRISE), which sampled frontal regions in the Northern Gulf of Mexico in 2021 using multiple platforms. Observations provide information about tracers (T/S and Oxygen) as well as dynamical quantities (vorticity, divergence), as the velocity gradient tensor could be estimated by running 2 ships in parallel tracks. Situations similar to the observed ones are modelled using realistic simulations with multiple nested levels where the final nest uses a non-hydrostatic code with 100 m horizontal grid-space. Finally, various diagnostics are applied to the model outputs to better characterize vertical exchanges, including Lagrangian diagnostics, as well as dynamical processes responsible, using in particular reduced-physics models.

Observations and model show intense diurnally modulated episodes of upwelling and downwelling happening in frontal areas, which are driven by the interaction of the diurnal land-sea breeze and the fronts. The vertical circulation is able to bring oxygen depleted bottom waters toward the surface, where they are mixed with surface waters and oxygenised. The detailed dynamical analysis further highlights that the intense convergence/divergence motions at the surface driving the vertical circulation are due to the winds, diurnally destabilising the front and mixing the geostrophic flow.

Overall this represents a very insightful study, based on an impressive dataset of in-situ observations and model outputs, and highlighting novel processes, which are of great interest in physical oceanography and biogeochemistry. The mechanisms highlighted here will likely apply to many other regions of the world and thus have a broad impact.

The paper is very well written, and details novel and important results, and the methodology is sound, therefore I recommend publication after the following minor concerns are addressed.

Minor comments:

- I am left wondering how crucial is the land-sea breeze to the sequence of events observed here. It is indeed shown in the paper that most frontal quantities (divergence,

up/downwelling) follow a diurnal/inertial cycle, but is it really the trigger or does it just modulate frontal circulations that would exist otherwise? In other words, could the same processes occur if the fronts were only forced by sustained (down-front) winds?

- Are there any metrics to quantify if frontal features have converged at the highest resolution used here (100 m) ? Some studies showed that the typical Rossby deformation radius over the shelf can be hundreds of meters or less (*e.g.*, Barkan et al. 2017).
- Fig 2D, 3A, 4C, Fig S5, : This colorbar can be confusing for colorblind people. Personally, I cannot distinguish between positive and negative values. Consider using a different colorscale (a regular blue-red like Fig. 2B is easier to distinguish).
- l. 79: Oxygen sections are shown for June 23 and June 25 in Fig. S3 and S4, but not for June 24. Why not show them for the same section (June 24) than the one shown in Fig. 2? The comparison would be more direct (or maybe it is somewhere in the manuscript and I missed it?).
- l. 217: Do the diurnal heating and cooling play any role here? Heat flux amplitudes in Fig. S7 are significant, though they do not appear to correlate well with viscosity variations. Is it because the viscosity variations due to diurnal heat fluxes are small compared to the ones due to the wind? Or is it because cooling phases happen during periods of restratification by the wind? By the way why is the heat flux minimum at 18:00?
- l. 297: Could the authors precise which turbulence closure scheme is selected among the different choices provided by GLS?
- Fig. 4, caption: "(f) same as D" → same as (E). I guess colors on panel F also correspond to Dissolved Oxygen?
- Fig. 6, caption: "generates and inertially-oscillating" → generates an inertially-oscillating
- References: Capital letters missing in several titles.

REFERENCES

- Barkan, R., J. C. McWilliams, A. Shchepetkin, L. Renault, A. Bracco, and J. Choi, 2017: Submesoscale dynamics in the northern Gulf of Mexico. Part I: Regional and seasonal characterization, and the role of river. *J. Phys. Oceanogr.*, **47**, 2325–2346.

Responses to Reviewers

We appreciate the time and effort that you dedicated to providing feedback on our manuscript and are grateful for the insightful comments on and valuable improvements to our paper. We have incorporated/addressed all the suggestions/comments. Please see below for a point-by-point response to the comments and concerns. The comments are in blue, and the responses are in black.

1 Responses to Reviewer #1

1.1 Responses to Major Comments

Major Comment (1):

The experimental design for the physical oceanography observations is elegant with the simultaneous coordinated measurements of two ship transects across a well-defined front. The physics identified along the frontal edge are thorough and groundbreaking. And, there was an obvious intrusion of bottom-water along the convergence towards the surface, and vice-versa, as identified from dissolved oxygen concentrations overlaid on the physics. The numerical simulations and explanations are well done and succinct. It should be emphasized in the text where simulations versus observations are presented.

Response:

Thanks for your insightful comments, which helped us effectively improve our manuscript. To emphasize where simulations versus observations are presented, we have clarified their placement in the manuscript at the end of the introduction. The result section is organized as follows: the first subsection (i.e., the SUNRISE Campaign 2021) shows the observational evidence of the vertical exchange to motivate the study, and the rest of the subsections focus on the simulations to demonstrate the underlying physics.

Major Comment (2):

More informal language, e.g., “indeed,” “so,” “breathe,” even if given single quotes, is up to the editors.

Response:

Thanks for the comments. We have replaced “breathe” with “pulses of upwelling” and replaced “indeed” with more formal wording.

Major Comment (3):

Study area figure 1: need to identify the Atchafalaya delta so that the reader (uninformed of nGoMx geography) understands the idea of the Mississippi/Atchafalaya

plume, which is then dropped to Mississippi River plume, possibly describe the two river plumes, relative inputs, general downcoast flow in June, in Figure 1 and associated text.

Response:

Thanks for the suggestions. We have denoted both the Atchafalaya and Mississippi deltas in Fig. 1A. We also have added the Atchafalaya river in the caption of Fig.1 and the text wherever the Mississippi river is mentioned.

Major Comment (4):

“These winds generate strong inertial oscillations, which set up a diurnally-pulsing vertical circulation at the fronts that draws bottom waters up to the surface mixed 21 layer.” This mechanism is verified by the dissolved oxygen concentrations.

Response:

We have revised the abstract, and this statement has been rephrased to clarify.

Major Comment (5):

L130, “near-surface low-oxygen water.” It would be helpful if the range of concentrations could be provided in the text. However, I can surmise that none of the values qualify for being “hypoxic,” given a range of 150-200 uM, which converts to approximately 80 to 100% saturation. This range of dissolved oxygen is in the upper range of values expected in this area in June, not near “hypoxic,” and the bottom-waters do not approach “hypoxia.”

Response:

Thanks for the comment. We have toned down the statements associated with hypoxia and replaced the “low oxygen” with “lower oxygen”.

Major Comment (6):

Figure 3H, indicates that the horizontal width of a ‘bore’ that reaches 30 m is less than 1 km and similar but not as well-defined to depths of 10-15 m.

Response:

The bore refers to the depression of isopycnals beneath the leading edge of the front as shown in Figs. 2F and 3G/H. Thanks for the feedback. We realized that ‘bore’ might be too jargon. We have replaced ‘bore’ with ‘depression of isopycnals’ and elaborated on that. Basically, the strongly convergent surface flow draws together isopycnal surfaces and induces downward motions which advect the buoyant surface water downwards and contribute to the depression of isopycnals beneath the leading edge of the front.

Major Comment (7):

Given the above results related to dissolved oxygen, it seems the intrusion/extrusion of different levels of dissolved oxygen is a minor process in the overall dissolved oxygen fields on this area of the Louisiana shelf. The last sentence of the abstract should be revised away from “an important role”; the data do not support the existing statement.

Response:

We have toned down the associated claims in the abstract and the main text.

Major Comment (8):

The diurnal cycles are the focus but there is no mention of the tidal cycles and their interactions, which should be present at this location. If they were part of the analysis, this needs to be stated, especially, if they were not identified in the physical environment. And, if not examined, this should also be stated.

Response:

The tides in this region is relatively weak. The wind-driven diurnally oscillating currents can reach 60 cm/s (*DiMarco et al.*, 2000), while the amplitude of the dominant, diurnal tidal currents (K1 and O1) is one order of magnitude smaller (see Tab.2 of *DiMarco and Reid* (1998)). We have clarified this in the introduction.

1.2 Responses to Minor Commentss

Minor Comments (1):

L14, suggest “support hypoxia formation and maintenance” instead of “condition”

Response:

Thanks for the comment. We have revised the sentence as suggested.

Minor Comments (2):

18, “...exchange at these fronts can be quite rapid and can lead to the oxygenation of bottom water” This reviewer will look for frontal vertical mixing within the depth gradient where hypoxia is likely to occur. Much of the plume front is over much deeper continental shelf areas than where hypoxia occurs.

21, these “breaths” the rate of oxygenation of the bottom waters is comparable to deoxygenation by the respiration of organic matter over the shelf. This reviewer will look for temporal respiration rates typical of shelf ecosystems comparable to those generated to the “breaths” observed by the authors.

Response:

Thanks for the comments. Regarding the location of the plume fronts, the fronts are actually active at both deeper shelf areas and shallower areas (see the rich field of plume fronts at the depths of < 20 m in Fig. 3A and 3B of the manuscript), suggesting that the vertical exchange can potentially occur at shallower areas as well. Also, the hypoxic zone can sometimes extend to deeper (> 25 m) shelf areas as shown in Fig. 1. Although 2019’s hypoxic zone in the Gulf of Mexico is the 8th largest measured in the past 33 years (based on the NOAA report), the process of the vertical exchange can potentially influence the hypoxic zone at deeper shelf areas once the hypoxic zone extends.

[REDACTED]

Figure 1: The hypoxic zone in the Gulf of Mexico measured in July, 2019. The figure is from the NOAA report at <https://www.noaa.gov/media-release/large-dead-zone-measured-in-gulf-of-mexico>.

Regarding the respiration rate, we are comparing it to the deoxygenation due to aerobic respiration of organic matter in July over the Texas-Louisiana shelf, which is about 1 mg/L/day reported by *Rabalais and Turner* (2019). This is comparable to the oxygenation rate (0.82 mg/L/day) estimated based on the Lagrangian particle tracking in our model, which solely accounts for the effect of the physical process. The other number that we could compare is the seasonal reduction of oxygen near the bottom from the transects averaged over 1985–2001, i.e., the bottom right panel in Fig.3 in *Rabalais and Turner* (2019), which is around 6 mg/L over 6 months, or 0.03 mg/L/day. This rate would be attributed to many different oxygenating and deoxygenating processes. We appreciate your feedback about the temporal respiration rates typical of shelf ecosystem. If you think that is crucial for this study, we are happy to conduct the associated analysis and include it in the manuscript if any relevant literature (which will be much appreciated) can be provided.

Minor Comments (3):

33-34, suggest connecting the surface waters with the bottom waters.

Response:

Thanks for the comment. We have revised the sentence as suggested.

Minor Comments (4):

L34-35, can evade the stratification barrier of the plume; meaning not clear

Response:

Due to the stratification barrier, vertical mixing is suppressed, and therefore diapycnal transport is suppressed. However, the slantwise fronts can provide conduits where the slantwise vertical motions at these fronts can bypass the stratification barrier by transporting water along their inclined isopycnals rather than penetrating the barrier via vertical mixing. We have clarified this in the introduction.

Minor Comments (5):

L 61, define "SUNRISE" here

Response:

It is defined in the introduction.

Minor Comments (6):

L 70, better description of Bore-like features

L73 more technical description of shock-like jumps

Response:

We have replaced 'Bore-like features' with 'depression of isopycnals' and elaborated on that. Basically, the strongly convergent surface flow draws together isopycnal surfaces and induces downward motions which advect the buoyant surface water downwards and contribute to the depression of isopycnals beneath the leading edge of the front.

Minor Comments (7):

L76, ADCP observations that substantiate the down-welling?

Response:

Unfortunately, we did not have robust measurements of vertical velocities during the cruise, which require more sophisticated instruments. However, through the coordination of two ships, we were able to estimate the convergence/divergence at the fronts using the horizontal velocities from the ADCPs. As shown in Fig. 2B, there is strong convergence on the leading edge of the front, indicating downwelling motions.

Minor Comments (8):

L79, suggest "cooler" water; the temperature range is 28.76 to 28.88 C in Figure 2 and 27.8 to 28.8 in Figure 4.

Response:

Thanks for the suggestion which we have incorporated. Fig. 2 is made based on the observations, while Fig. 4 is made based on the simulation. Since the observations and simulations are in different years, the temperature ranges are slightly different. In order to better visualize, the colorbars in these two figures are adjusted to highlight the subduction and intrusion.

Minor Comments (9):

L102, suggest something other than “nose.” Possible, peak, apex...

Response:

We have replaced “nose” with “leading edge”.

Minor Comments (10):

L108, is largely dependent on salinity for most of the year [addition: with a lower-water column thermocline contributing to stratification differences in the summer].

Response:

Thanks. We have added the statement as suggested.

Minor Comments (11):

L115,... indicating rapid vertical transport... However, as noted in other comments, these intrusions are of limited geographical expanse and ephemeral.

Response:

Regarding the geographical expanse, although the fronts look narrow at the surface with the width of $O(1)$ km, the interior intrusion and subduction (that are driven by the convergence/divergence at the fronts) expand a larger area with the width of $O(10)$ km (see Fig. 4A/B of the manuscript). Given that the surface fronts cover about 5-10% of the shelf region (based on the percent area with surface normalized relative vorticity greater than one) and the spatial scale of the intrusion/subduction is one order of magnitude larger than the fronts, the geographical expanse of the intrusion/subduction is not insignificant. Regarding the temporal characteristics, as shown in Fig. 1C, the intrusion diurnally pulses, which would persist if the diurnal land-sea breeze sustains (over the Texas-Louisiana shelf, diurnal land-sea breeze is prominent during the summer from June to August (*DiMarco et al.*, 2000)). But more importantly, these diurnal pulses lead to irreversible changes in the properties of the water (e.g. oxygen, PV, etc.) which persist even after the diurnal-varying circulation subsides.

Minor Comments (12):

L123,... particularly evident in the oxygen field... As noted in other comments the dissolved oxygen range is minimal and not “hypoxic.”

Response:

We have clarified the statement according to the comment.

Minor Comments (13):

L208, suggest “denser”

Response:

We used “dense side” and “light side”, for brevity, to distinguish the denser side and the lighter side of the front. But we appreciate your feedback.

Minor Comments (14):

L229, should italicize Sargassum

Response:

Thanks. We have italicized Sargassum.

Minor Comments (15):

L251-252, statements such as "...could serve as a significant ventilation mechanism for the bottom waters on the shelf and play an important role in the evolution of dead zones in the region." are not supported by the data or the simulations. The data in this manuscript represent a period in early June, and the low-oxygen area persists substantially from June – August or September, under strengthened stratification and long periods of no re-aeration. Suggest "dynamics" instead of "evolution," and "potential" for "important role."

L252, ... introduction of low-oxygen, nutrient-rich water into the photic zone may play an important role in... "agree with the concept, but it needs proof. Suggested wording "introduction of nutrient-rich water associated with low oxygen..." It is not the low oxygen that is important in the process but the nutrient-rich bottom waters.

L254, ... Frontal processes create strong temporal variability... As noted the temporal duration is minimal, and the dissolved oxygen differences are also minimal. Suggest that this finding be reworded.

Response:

We have rephrased the findings of this study following the suggestions of the reviewer.

Minor Comments (16):

L255, suggest that the wording should be "...to be an environment in which hypoxia..."

Response:

Thanks. We have incorporated this suggestion.

Minor Comments (17):

L256, suggest ...remains unclear...

Response:

Thanks. We have incorporated this suggestion.

Minor Comments (18):

L263, the idea of tidal processes is just now being introduced. Suggestions made earlier about tidal cycles. Tidal advection is more prominent where the depth gradient is steeper, not in the small gradient of the study area.

Response:

Thanks for the comment. We have stated in the introduction that the tides in this region is relatively weak. The wind-driven diurnally oscillating currents can reach 60 cm/s (*DiMarco et al.*, 2000), while the amplitude of the dominant, diurnal tidal currents (K1 and O1) is one order of magnitude smaller (see Tab.2 of *DiMarco and Reid* (1998)).

Minor Comments (19):

L266, Information about the frequency of frontal passages in the summer in the nGoMex are available in the literature and should be included.

Response:

Thanks. We have added references there.

Minor Comments (20):

L275, delete “fulfill.”

Response:

Deleted.

Minor Comments (21):

L300, the surface waters may be super-saturated in dissolved oxygen at this time of year, despite the warm water temperatures.

Response:

Thanks for the comment. It is an interesting point that the surface waters may be super-saturated in dissolved oxygen at this time of year. It would create sharper gradients of oxygen at the surface mixed layer where the upwelled bottom water enters and therefore yield higher oxygenation rates. It is definitely something we should consider and improve in the future research.

Minor Comments (22):

References: please make uniform according to the journal format, e.g., L354, capitalize journal name, here and elsewhere; L356, insert . . . , USA after PNAS;

Response:

Thanks for the suggestions. We have capitalized the journal names and corrected the typos.

Minor Comments (23):

L429, should the URL link be provided here?

Response:

We believe that the journal system will generate a URL link for the supplementary materials when the manuscript becomes online.

Graphics Comments:

- Excellent graphics, but clarity on some could be improved.
- Figure 3. Can a more technical word than “bore” be used, e.g., “intrusion”?
- The infographic (Fig. 6) is well done and informative. I suggest an arrow for North be included.
- Figure 6 caption: Schematic of the key processes contributing to the vertical circulation. A diurnal land-sea breeze wind-driven generates [and] inertially-oscillating surface flow. Should the “and” be “an”?
- Figure 6 caption, ... a bottom oxygen-deficient water mass can “breathe” through a channel reaching the surface mixed layer, where oxygenation can be stimulated by various physical and biochemical processes.
- Suggest, ...can insert lower dissolved oxygen waters into the surface mixed layer through. ...
- Delete ...stimulated by various physical [delete and biochemical] processes Bio-geochemistry is not included in this manuscript.

Responses:

Thanks for the suggestions. Regarding Fig. 3, we have replaced “Bore” with “Depression” because the isopycnals are depressed due to the downward advection of buoyant water induced by the convergence near the surface. Regarding Fig. 6, we think that the geostrophic orientation is not essential to illustrate the key elements of the theory, so we decided not to include an arrow for North for clarity. But we appreciate your feedback. Other suggestions have been integrated into the figure and the caption.

2 Responses to Reviewer #2

2.1 Responses to Major Comments

Major Comment:

In the article Rapid Vertical Exchange at Fronts in the Northern Gulf of Mexico by Lixin Qu et al, the authors investigate the vertical motions happening in frontal regions forced by the land-sea breeze over the Northern Gulf of Mexico and highlight potentially large impacts for oxygenation of bottom waters.

The study uses a unique set of observations, state-of-the-art numerical simulations, and elaborated diagnostics to characterize vertical motions happening at fronts and the mechanisms driving them. Observations come from a dedicated cruise (SUNRISE), which sampled frontal regions in the Northern Gulf of Mexico in 2021 using multiple platforms. Observations provide information about tracers (T/S and Oxygen) as well as dynamical quantities (vorticity, divergence), as the velocity gradient tensor could be estimated by running 2 ships in parallel tracks. Situations similar to the observed ones are modelled using realistic simulations with multiple nested levels where the final nest uses a non-hydrostatic code with 100 m horizontal grid-space. Finally, various diagnostics are applied to the model outputs to better characterize vertical exchanges, including Lagrangian diagnostics, as well as dynamical processes responsible, using in particular reduced-physics models.

Observations and model show intense diurnally modulated episodes of upwelling and downwelling happening in frontal areas, which are driven by the interaction of the diurnal land-sea breeze and the fronts. The vertical circulation is able to bring oxygen depleted bottom waters toward the surface, where they are mixed with surface waters and oxygenated. The detailed dynamical analysis further highlights that the intense convergence/divergence motions at the surface driving the vertical circulation are due to the winds, diurnally destabilising the front and mixing the geostrophic flow.

Overall this represents a very insightful study, based on an impressive dataset of in-situ observations and model outputs, and highlighting novel processes, which are of great interest in physical oceanography and biogeochemistry. The mechanisms highlighted here will likely apply to many other regions of the world and thus have a broad impact.

The paper is very well written, and details novel and important results, and the methodology is sound, therefore I recommend publication after the following minor concerns are addressed.

Response:

We appreciate the time and effort that you dedicated to providing the insightful comments on and valuable improvements to our paper.

2.2 Responses to Minor Comments

Minor Comment (1):

I am left wondering how crucial is the land-sea breeze to the sequence of events observed here. It is indeed shown in the paper that most frontal quantities (divergence, up/downwelling) follow a diurnal/inertial cycle, but is it really the trigger or does it just modulate frontal circulations that would exist otherwise? In other words, could the same processes occur if the fronts were only forced by sustained (down-front) winds?

Response:

If the fronts were only forced by sustained, down-front winds, symmetric instabilities would be triggered which resemble a pattern of slantwise convection but do not inertially pulse. This scenario is essentially associated with the wind-forced symmetric instability which has been discussed by *Thomas and Taylor (2010)*. In our theory, the inertially oscillating flow acting over a front, at certain phases, advects dense water over light and triggers the inertially modulated frontogenesis including the sequence of processes as schematized in Fig.6 of the manuscript. Why does the flow inertially oscillate? That is because of the land-sea breeze which is near-resonant with the local inertial period and therefore resonantly forces the near-inertial flow. Consequently, the land-sea breeze is crucial to the sequence of events discussed in the study.

Minor Comment (2):

Are there any metrics to quantify if frontal features have converged at the highest resolution used here (100 m) ? Some studies showed that the typical Rossby deformation radius over the shelf can be hundreds of meters or less (e.g., *Barkan et al. (2017)*).

Response:

We had a nested CROCO simulation with an even higher resolution of 25 m, which is termed TXLA-L4. Fig. 2 shows the intercomparison of the temperature along the same transect between TXLA-L3 (with a resolution of 100 m) and TXLA-L4. These two simulations exhibit nearly identical patterns of the subduction of surface warm water and the upwelling of bottom cooler water. Also, both simulations reproduce the observations from the cruise including the density field and the temperature field (Fig. 2). Based on these consistencies, we believe that the frontal features have converged in TXLA-L3 which is used in the manuscript.

Minor Comment (3):

Fig 2D, 3A, 4C, Fig S5, : This colorbar can be confusing for colorblind people. Personally, I cannot distinguish between positive and negative values. Consider using a different colorscale (a regular blue-red like Fig. 2B is easier to distinguish).

Response:

Thanks for the feedback. We have changed the colormap in Fig. 2D, 3A, 4C, 5B, and Fig. S5 to the regular blue-red as suggested.

Figure 2: Intercomparison of temperature between the observation and simulations. The left panel shows the observation of temperature on the R/V Pelican along the transect shown in Fig. 2A of the manuscript. The middle and right panels show the simulated temperature from the TXLA-L3 and TXLA-L4, respectively, along the cross-front transect shown in Fig. 3E of the manuscript. TXLA-L3 has the horizontal resolution of 100 m. TXLA-L4 has the horizontal resolution of 25 m.

Minor Comment (4):

1. 79: Oxygen sections are shown for June 23 and June 25 in Fig. S3 and S4, but not for June 24. Why not show them for the same section (June 24) than the one shown in Fig. 2? The comparison would be more direct (or maybe it is somewhere in the manuscript and I missed it?).

Response:

Unfortunately, we do not have oxygen observations on the sections of June 24 shown in Fig. 2, because the oxygen sensor and the vertical microstructure profilers were irregularly swapped among sections. The oxygen sections on June 23 and 25 shown in Fig. S3 and S4 are the most relevant sections in time that we can use to establish the comparison.

Minor Comment (5):

1. 217: Do the diurnal heating and cooling play any role here? Heat flux amplitudes in Fig. S7 are significant, though they do not appear to correlate well with viscosity variations. Is it because the viscosity variations due to diurnal heat fluxes are small compared to the ones due to the wind? Or is it because cooling phases happen during periods of restratification by the wind? By the way why is the heat flux minimum at 18:00?

Response:

The diurnal heat flux plays a less important role than the Ekman buoyancy flux. As shown in Fig. S7, the amplitude of the heat flux is one or two orders of magnitude smaller than the one of the Ekman buoyancy flux. Consequently, the viscosity correlates with the Ekman buoyancy flux much better than the heat flux.

The time is UTC time. The heat flux minimum at 18:00 UTC is corresponding to the heat gain maximum happening at 12:00 local time (the local time zone is UTC-6h). We have clarified this in the caption of Fig. S7 and other figures. Thanks for the insight.

Minor Comment (6):

l. 297: Could the authors precise which turbulence closure scheme is selected among the different choices provided by GLS?

Response:

Thanks for the comment. $k - \epsilon$ is selected. We have clarified this in the method section.

Minor Comment (7):

Fig. 4, caption: "(f) same as D" → same as (E). I guess colors on panel F also correspond to Dissolved Oxygen?

Response:

Thanks for the comment. We have corrected the typo in the caption. Indeed, the colors on panel F also correspond to dissolved oxygen.

Minor Comment (8):

Fig. 6, caption: "generates and inertially-oscillating" → generates an inertially-oscillating

Response:

Thanks for the comment. We have corrected the typo in the caption.

Minor Comment (9):

References: Capital letters missing in several titles.

Response:

Thanks for the comment. We have fixed this issue.

References

- Barkan, R., J. C. McWilliams, A. F. Shchepetkin, M. J. Molemaker, L. Renault, A. Bracco, and J. Choi (2017), Submesoscale dynamics in the northern gulf of mexico. part i: Regional and seasonal characterization and the role of river outflow, *Journal of Physical Oceanography*, 47(9), 2325–2346.
- DiMarco, S. F., and R. O. Reid (1998), Characterization of the principal tidal current constituents on the Texas-Louisiana Shelf, *J. Geophys. Res.*, 103(2), 3093–3110.
- DiMarco, S. F., M. K. Howard, and R. O. Reid (2000), Seasonal variation of wind-driven diurnal current cycling on the Texas-Louisiana Continental Shelf, *Geophys. Res. Lett.*, 27(7), 1017–1020.
- Rabalais, N. N., and R. E. Turner (2019), Gulf of mexico hypoxia: Past, present, and future, *Limnology and Oceanography Bulletin*, 28(4), 117–124.
- Thomas, L. N., and J. R. Taylor (2010), Reduction of the usable wind-work on the general circulation by forced symmetric instability, *Geophysical Research Letters*, 37(18).